# Effect of Post-COVID-19 on Brain Volume and Glucose Metabolism: Influence of Time Since Infection and Fatigue Status

**DOI:** 10.3390/brainsci13040675

**Published:** 2023-04-18

**Authors:** Justin R. Deters, Alexandra C. Fietsam, Phillip E. Gander, Laura L. Boles Ponto, Thorsten Rudroff

**Affiliations:** 1Department of Health and Human Physiology, University of Iowa, Iowa City, IA 52242, USA; justin-deters@uiowa.edu (J.R.D.); alexandra-fietsam@uiowa.edu (A.C.F.); 2Department of Radiology, University of Iowa Hospitals and Clinics, Iowa City, IA 52242, USA; phillip-gander@uiowa.edu (P.E.G.); laura-ponto@uiowa.edu (L.L.B.P.)

**Keywords:** post-COVID-19, FDG-PET, brain volume, fatigue, neuroimaging in post-COVID-19

## Abstract

Post-COVID-19 syndrome (PCS) fatigue is typically most severe <6 months post-infection. Combining magnetic resonance imaging (MRI) and positron emission tomography (PET) imaging with the glucose analog [^18^F]-Fluorodeoxyglucose (FDG) provides a comprehensive overview of the effects of PCS on regional brain volumes and metabolism, respectively. The primary purpose of this exploratory study was to investigate differences in MRI/PET outcomes between people < 6 months (N = 18, 11 female) and > 6 months (N = 15, 6 female) after COVID-19. The secondary purpose was to assess if any differences in MRI/PET outcomes were associated with fatigue symptoms. Subjects > 6 months showed smaller volumes in the putamen, pallidum, and thalamus compared to subjects < 6 months. In subjects > 6 months, fatigued subjects had smaller volumes in frontal areas compared to non-fatigued subjects. Moreover, worse fatigue was associated with smaller volumes in several frontal areas in subjects > 6 months. The results revealed no brain metabolism differences between subjects > 6 and < 6 months. However, both groups exhibited both regional hypo- and hypermetabolism compared to a normative database. These results suggest that PCS may alter regional brain volumes but not metabolism in people > 6 months, particularly those experiencing fatigue symptoms.

## 1. Introduction

The COVID-19 pandemic, caused by the SARS-CoV-2 virus, has, as of March 2023, resulting in over 750 million confirmed cases worldwide, including over 100 million cases and >1 million deaths in the United States (World Health Organization; WHO). Initially, the focus of physicians and researchers was on acute symptoms and care. However, three years into the pandemic, more focus has been put on the long-term outcomes of COVID-19 infection [1,2], particularly the prevalence of post-COVID-19 syndrome (PCS) symptoms. PCS is characterized as symptoms that last for at least two months after an acute COVID-19 illness and cannot be explained by an alternate diagnosis (WHO). Common symptoms include olfactory dysfunction (anosmia/parosmia/dysosmia), cognitive impairment, and, most commonly, fatigue [3,4,5]. Fatigue is a complex, multi-faceted phenomenon arising from various factors [5] and typically consists of feelings of tiredness and decreases in energy, motivation, and concentration, which can only be measured by self-reported questionnaires. On the contrary, fatigability is a measure of physical or cognitive work capacity. Particularly, perceived fatigue (i.e., feelings of tiredness and lack of energy) and fatigability (i.e., a subjective estimate of future work capacity) have been shown to be more prevalent than objective fatigue (i.e., an objectively measured decrease in performance on a specific task) in people with PCS [6]. These symptoms have been reported to last over 6 months in some patients [7], with 20% of subjects reporting fatigue even 1-year post-infection [8]. However, the most severe symptoms are commonly reported within the first 6 months post-infection [9].

It is still widely debated whether the virus can enter the brain and if the reported neurological symptoms (e.g., fatigue, cognitive impairment, dysosmia) are direct or indirect manifestations of the infection [10,11]. Regardless of the source (i.e., direct or indirect invasion), it has been proposed that brain morphometrical alterations could arise from several factors, such as decreased sensory input (e.g., dysosmia) or chronic inflammation [10,12]. Recent studies have investigated morphometric changes over time in the brain after acute COVID-19 and reported mixed results [1,2,12,13]. Specifically, Douaud et al. showed greater rates of atrophy of certain olfactory and limbic regions in PCS patients ~5 months post-infection compared to healthy controls, but no association between time since infection and brain volume was demonstrated [12]. Similarly, Tian et al. investigated changes in brain morphometry 3 months and 10 months after COVID-19 infection and found atrophy of the left putamen and thalamus after 3 months, which was not recovered after 10 months, and atrophy of the right putamen and nucleus accumbens after 10 months, which was not present at the 3-month timepoint [2]. Moreover, Besteher et al. found larger volumes in PCS patients with neuropsychiatric symptoms (e.g., depression and cognitive impairment) an average of 8 months after COVID-19 infection [13]. However, despite variability in brain volumes at different time points, both Tian et al. and Besteher et al. found inverse associations between volume and time, with volume decreasing over time [2,13]. Additionally, Du et al. compared brain regional volumes at 1 and 2 years after COVID-19 infection, finding persistently smaller volumes in deeper brain regions but recovered volumes in frontal and temporal areas at year 2 compared to year 1 [1]. These studies suggest that acute COVID-19 may influence cerebral volumes up to 2 years post-infection [1], although more research is needed to clarify some of the conflicting reports and to fully delineate the timeline of possible effects. The inconsistency in results may be due to the lack of investigations focusing on the effects of PCS after mild (i.e., not hospitalized) COVID-19 infection on brain morphometry over time, especially comparing post-acute (1–6 months post-infection) phases versus chronic (>6 months post-infection).

A metabolic neuroimaging study combined with structural imaging can help develop a comprehensive overview of brain alterations after acute COVID-19. [^18^F]Fluorodeoxyglucose (FDG) is a radioactive tracer used in positron emission tomography imaging (PET) that is an analog of glucose, making it a useful tool for the evaluation of brain metabolism [14,15]. Meyer and colleagues recently provided an overview of the FDG-PET literature in PCS and found conflicting results [16]. Localized regional hypometabolism was reported in adults with PCS compared to healthy controls [17], in melanoma patients [18], and in children with PCS compared to healthy controls [19]. In contrast, one study found no evidence of metabolic alterations in people with PCS compared to healthy controls [20], while Martini et al. found hypometabolism in various regions (e.g., frontal cortex, insula, and thalamus) and hypermetabolism in the brainstem, cerebellum, hippocampus, and amygdala [10]. While the hypometabolism recovered by the 7–9 month timeframe, hypermetabolism persisted [10]. Similar to the structural imaging results in people with PCS, these studies imply location-specific and time-dependent effects on brain metabolism [10] and that the presence or absence of fatigue may influence these findings [20]. The reported results may be inconsistent between studies due to various design issues, such as the population and timeframe being investigated (i.e., pediatric vs. adult, active infection vs. directly after infection, acute infection vs. PCS), and the choice of control or comparator populations (no COVID-19 diagnosis vs. recovered COVID-19 patients). However, the results of these studies suggest that there are pathological processes ongoing during an acute infection or as secondary to the initial infection that may alter brain metabolism, which warrants further investigation. For a more comprehensive review of imaging in PCS, see Okrzeja et al. [21].

In performing both structural and metabolic imaging to assess the brains of people at different time points after acute mild COVID-19, the same study could aid in the understanding of PCS. Important considerations that may influence these outcomes include the presence and severity of post-COVID-19 symptoms, particularly fatigue [4]. The primary purpose of this exploratory study was to investigate differences in brain structure and metabolism between people after acute, mild COVID-19 infection during two timeframes (i.e., <6 months and >6 months post-infection). The secondary purpose was to determine if the presence of observed structural or metabolic differences could explain, at least in part, the presence of fatigue symptoms. We hypothesized that subjects < 6 months post-infection would show altered brain volumes and glucose metabolism in various brain regions compared to subjects > 6 months post-infection. Furthermore, the alterations in volume and glucose metabolism can explain, at least in part, the presence or absence of fatigue symptoms [20].

## 2. Materials and Methods

### 2.1. Subjects

The subjects were recruited via mass email at the University of Iowa and the University of Iowa Hospitals and Clinics (>35,000 recipients). 190 potential subjects were identified, and thirty-three subjects (17 female) who were previously diagnosed with COVID-19 were eventually enrolled between January 2021 and June 2022. Inclusion criteria were: (1) being between the ages of 18–80, (2) meeting CDC guidelines for discontinuation of home isolation and being at least 6 weeks post-quarantine; (3) having the ability to read, write, speak, and understand English, as well as comprehension of the protocol; (4) confirming the COVID-19 diagnosis via medical record; and (5) experiencing or not experiencing PCS symptoms (depending on group assignment), according to the Chalder Fatigue Scale CFQ-11 (PCS fatigue ≥5 [6,22]; completed via email or phone). Exclusion criteria were: (1) history of traumatic brain injury or hydrocephalus; (2) pregnancy; (3) the presence of any medical condition that may exacerbate any post-COVID-19 symptoms (e.g., major depressive disorder, anxiety disorder); and (4) hospitalization due to COVID-19 infection. This study was approved by the University of Iowa Institutional Review Board (IRB #202009381) and conducted in accordance with the Declaration of Helsinki.

### 2.2. Experimental Protocol

This protocol consisted of a single visit to the University of Iowa Hospitals and Clinics. Subjects provided consent, and then information about the diagnosis date and vaccine status was collected. Then, subjects underwent a positron emission tomography (PET) scan followed by a magnetic resonance imaging (MRI) scan.

### 2.3. Fatigue and Fatigability Assessments

The Fatigue Assessment Scale (FAS) is a subjective evaluation of perceptions of fatigue that contains 10 different statements [23]. The subject is asked to respond to each statement with a number from 1–5 about how much they felt it applied to them, with 1 being “never” and 5 being “always”. Scores on this scale range from 10 (no effects of fatigue) to 50 (very strong effects of fatigue).

The Fatigue Severity Scale (FSS) consists of nine statements to evaluate perceived fatigue [6], each rated on a 7-point scale with higher numbers representing higher agreement with the statement. The scores of each question are averaged to give an overall score between 1 and 7, with a score ≥4 indicating a clinically significant level of fatigue [24].

### 2.4. Imaging Acquisition

Prior to the imaging, subjects were required to fast for 6 h and have a blood glucose level ≤ 200 mg/dL to ensure proper uptake of the glucose tracer. Before imaging, 5 mCi ± 10% FDG was injected intravenously, and tracer uptake was completed while the subject lay supine in a dark room with their eyes open and their ears unplugged with no external stimuli. Imaging paralleled that employed in the Alzheimer’s Disease Neuroimaging Initiative (ADNI) protocols (i.e., 30 min of imaging commencing at 30 min post-injection in 5 min frames) [25]. Attenuation correction computed tomography (CT) and PET emission imaging were performed on a GE Discovery MI PET/CT (GE Healthcare, Chicago, IL, USA). The PET images were reconstructed using six iterations and 16 subsets (VUE Point HD method) with a 25.6 cm field of view and 192 × 192 matrix size, corrected for motion across acquisition frames, and then averaged across the 30 min. The MRI was captured on a GE SIGNA Premier 3T scanner (GE Healthcare, Waukesha, WI, USA) with a 48-channel head coil. A volumetric coronal T1 MP-RAGE (TI = 900 ms, TE = 2.952 ms, TR = 7.24 s, flip angle = 8°, FOV = 256 × 256 mm, matrix = 256 × 256, bandwidth = 244 Hz/pixel) scan was acquired with a 1.0 mm isotropic spatial resolution.

### 2.5. MRI Analysis

A volume analysis was completed by generating volumes of interest (VOIs) using each subject’s T1 MRI image. VOIs were generated with the NEURO tool of PMOD (PMOD Biomedical Image Quantification, Version 4.0, Zurich, Switzerland) and constructed via the Hammer’s Atlas [26]. The sample size of this study relative to other MRI comparisons is small [27]. Therefore, to facilitate comparisons between subjects and to help account for confounding variables such as sex, height, and body weight, the volume of each region was normalized to the global brain volume (Relative Regional Volume; RRV; e.g., 0.5 represents a structure that is 0.5% of the global volume). Thus, this number represents the distribution of the volume within a certain structure, not the absolute size of the region. This was performed because a comparison of absolute sizes with a small sample size would be highly susceptible to potential confounders.

### 2.6. PET Analysis

The FDG-PET image was co-registered with the subject’s T1 MRI, and the same VOIs were used as for the RRV analysis. Standardized uptake values (SUV) were calculated for each region of interest (ROI; normalized to subject weight) as well as a volume-weighted global average SUV. SUV values were then compared between groups. Additionally, relative regional metabolism (RRM) was calculated for each VOI by normalizing each individual SUV to the global SUV. Therefore, RRM is a comparison of the distribution of metabolism, helping to account for the small sample size and potential for confounding variables such as blood glucose levels, age, sex, height, and weight. A value of 1.2 is interpreted as having a 20% higher metabolism than the global average.

### 2.7. Normative Database Comparison

In addition, an analysis was performed using the MIM software suite (v7.0.5, MIM Software Inc., Beachwood, OH, USA) to provide a normative comparison with a database of 43 healthy subjects [^19^F] aged 40–80 years [15]). Z-scores were calculated by comparing values for each subject’s summed FDG image across template-derived regions to the proprietary normative database of FDG data, using whole-brain as the reference tissue. A time group (<6 months and >6 months) comparison was also performed to determine if differences emerge at various time points after acute COVID-19. A z-score for a particular region ≥1.96 or ≤−1.96 was considered to represent hypermetabolism or hypometabolism, respectively, compared to the normative database.

### 2.8. Statistical Analysis

The normality of all data was assessed with the Shapiro-Wilk tests, and sphericity was assessed with Mauchly’s test of sphericity. Age, height, and weight of the subjects were compared between the groups (<6 months post-infection and >6 months post-infection; <6 months fatigued vs. <6 months non-fatigued; >6 months fatigued vs. >6 months non-fatigued) with independent *t*-tests and sex distribution was assessed with chi-square analysis. Based on recent literature showing unique alterations at different time points after COVID-19 diagnosis [1,2,12], our comparison focused on the frontal lobe, basal nuclei, and limbic system regions (36 ROIs in Hammer’s Atlas; 27 ROIs in MIM). These regions were selected due to their role in similar diseases (multiple sclerosis [28,29], chronic fatigue syndrome [30]), and their contribution to common PCS symptoms such as dysosmia [31,32] and ageusia [33]. SUV, RRM, and RRV results between the groups were assessed with unpaired *t*-tests. To investigate the potential role of fatigue (the most prevalent symptom in post-COVID-19 syndrome [4]), unpaired *t*-tests comparing fatigued vs. non-fatigued subjects were only performed if the time comparison (<6 months post-infection vs. >6 months post-infection) demonstrated significant differences. For any regions with significant differences between fatigued and non-fatigued groups, Pearson’s correlations were performed between fatigue scales and the relevant imaging outcome to further clarify the potential effects of fatigue.

## 3. Results

### 3.1. Subject Characteristics

All subjects met the inclusion and exclusion criteria and completed the PET and MRI scans. Subject demographics for each group (<6 months and >6 months) are listed below in Table 1a, and further division by fatigue status is summarized in Table 1b. All the data met the requirements for normality, so parametric testing was used. The chi-square test showed no difference in the distribution of sex between groups (*p* = 0.49) and *t*-tests comparing age, height, and weight between groups were all non-significant (*p* ≥ 0.23), except for the >6 month fatigued vs. non-fatigued age comparison (*p* = 0.02), which revealed that non-fatigued subjects were younger. However, the average age of both groups is below 40, which is when progressive decreases in brain volume are commonly reported [34], and only 3 subjects were >40 years in the fatigued group, with the oldest being 54. Additionally, dividing by vaccination status did not present large enough samples to do comparisons. However, a previous systematic review suggested a possible protective effect of vaccination against developing PCS [35], therefore we thought it pertinent to include vaccination status information for further context about our sample.

### 3.2. Structural Analysis

#### 3.2.1. Relative Regional Volume Comparison: <6 Months Post-Infection vs. >6 Months Post-Infection

The results of the unpaired *t*-tests showed that subjects who were >6 months post-infection had smaller RRVs in the bilateral putamen, pallidum, and left and right thalamus compared to subjects < 6 months post-infection. Moreover, the right cuneus showed a significantly larger RRV in subjects > 6 months post-infection. These results are summarized in Table 2a, and a representative image is shown in Figure 1.

#### 3.2.2. Relative Regional Volume Comparison: Fatigued vs. Non-Fatigued in >6 Months Post-Infection Group

Differences in RRV were demonstrated between the >6 months post-infection and <6 months post-infection groups. Therefore, further analysis was carried out to determine if fatigue status influenced these differences. The unpaired *t*-tests comparing fatigued vs. non-fatigued subjects > 6 months post-infection revealed smaller relative volumes in the fatigued group compared to the non-fatigued group in the following regions: bilateral middle frontal gyri and straight frontal gyri; left anterior orbital gyrus; left inferior frontal gyrus; bilateral middle orbital gyri; right superior posterior temporal gyri; and right anterior cingulate gyrus. Frontal region results are shown in Figure 2.

#### 3.2.3. Relative Regional Volume Comparison: Fatigued vs. Non-Fatigued in <6 Months Post-Infection

In addition, smaller RRVs were seen in fatigued subjects in the left caudate nucleus, right nucleus accumbens, and left substantia nigra compared to non-fatigued subjects. The results of all fatigued vs. non-fatigued comparisons are summarized in Table 2b above.

#### 3.2.4. Relative Regional Volume and FSS Correlations

Pearson’s correlations in subjects > 6 months post-infection revealed significant negative associations with FSS score (i.e., a smaller RRV was associated with a higher FSS score/worse fatigue) in several brain regions: the bilateral middle frontal gyri, the straight frontal gyri, and the left inferior frontal gyrus. The correlations were not significant in the following regions: the left anterior orbital gyrus, the bilateral middle orbital gyri, the right superior posterior temporal gyrus, and the right anterior cingulate gyrus.

Pearson’s correlations in subjects < 6 months post-infection revealed significant negative associations between FSS and RRV, with smaller RRV associated with worse FSS scores in the right nucleus accumbens. There were no significant associations in the left caudate nucleus or left substantia nigra.

#### 3.2.5. Relative Regional Volumes and FAS Correlations

Pearson’s correlations for subjects > 6 months post-infection demonstrated that RRV was negatively associated with a higher FAS score (worse perceived fatigability) in several regions: the bilateral middle frontal gyri and straight frontal gyri; the left inferior frontal gyrus; the bilateral middle orbital gyri; and the right anterior cingulate gyrus. No significant associations were present in the left anterior orbital gyrus or right superior posterior temporal gyrus.

In the <6 month post-infection group, a smaller RRV was negatively associated with a higher FAS score in the right nucleus accumbens. No significant associations were found in the left caudate nucleus or left substantia nigra. All correlation results are summarized in Table 3 and visualized in Figure 3.

### 3.3. Metabolic Analysis

#### 3.3.1. SUV and Relative Regional Metabolism

Unpaired *t*-tests between <6 months and >6 months for SUVs and relative regional metabolism revealed no significant differences in any brain region (all *p* ≥ 0.12).

#### 3.3.2. Normative Database Comparison

Subjects < 6 months post-infection demonstrated hypometabolism relative to a healthy subject database in the pallidum (globus pallidus; average z = −2.35; 11/18 subjects) and hypermetabolism in the lateral orbital gyrus (average z = 2.1). Subjects > 6 months post-infection showed hypometabolism in the pallidum (average z = −2.38; 12/15 subjects), along with hypermetabolism in the caudate nucleus (average z = 1.98; 7/15 subjects).

## 4. Discussion

The main finding of this exploratory investigation was that individuals > 6 months post-infection had smaller relative volumes in various brain regions compared to people < 6 months post-infection. Moreover, fatigued individuals had smaller relative volumes than non-fatigued individuals in the >6 months post-infection group. Furthermore, fatigue and perceived fatigability were also associated with smaller relative volumes, particularly in frontal lobe areas. Interestingly, despite structural differences, no differences in cerebral glucose metabolism were demonstrated between groups. Finally, subjects < 6 months post-infection had both hypo- and hypermetabolism in certain regions, and the >6 months group exhibited hypometabolism when compared to a normative FDG-PET database. Table 3 summarizes our results compared to previous studies.

The results of the current study revealed that people > 6 months after acute COVID-19 infection showed lower relative volume in the bilateral putamen, pallidum, and thalamus compared to people < 6 months post-infection. This is consistent with a recent report of autopsies performed in patients who died with COVID-19 that showed detectable levels of SARS-CoV-2 RNA in the thalamus, hypothalamus, cerebral cortex, and basal ganglia immediately after infection and more than 31 days post-infection [37]. Importantly, these regions may play a role in common PCS symptoms, such as fatigue and dysosmia [4,38]. The pallidum and putamen are part of the basal nuclei and are functionally connected to the thalamus [39]. The basal nuclei and thalamus are known key contributors to fatigue in various neurological populations such as multiple sclerosis (MS), Parkinson’s disease, Alzheimer’s disease (AD), chronic fatigue syndrome, stroke, and older adults [27,40,41,42,43,44,45]. Additionally, the thalamus is part of the secondary olfactory cortex, which has shown decreased volume in people after COVID-19 infection, potentially due to olfactory loss/dysfunction [12,31,46]. Moreover, some patients do not fully recover or have altered olfaction after COVID-19 infection, which may result in atrophy of associated brain regions over time [12,46,47]. Therefore, the structural alterations seen in the thalamus, putamen, and pallidum in the current study may arise from direct invasion of the virus into the CNS [36] or secondary mechanisms related to infection, such as the presence of fatigue and/or decreased sensory input.

The results of the current study are in line with the findings of Díez-Cirarda et al., who reported decreased regional gray matter volumes (frontal gyrus, temporal, and limbic areas) in people around 11 months post-infection [48]. However, these findings are contradictory to several recent reports. Increased volumes up to 8 months post-infection have been reported [13,49,50], while other studies showed no long-term changes in regional volumes [2,12]. These conflicting results may be due to variations in age, acute COVID-19 severity (e.g., hospitalization status, need for a ventilator), and time since diagnosis. Specifically, several studies evaluated older populations on average (≥ 44 years vs. ≥ 30.40 years; [48,49,50]) and included hospitalized subjects, while our subjects all reported mild cases (i.e., no hospitalization or ventilator support). Moreover, Douaud and colleagues noted that less than 20% of their subjects were more than 6 months post-COVID-19, making it difficult to draw conclusions about the effect of time on this population. Our results posit that time is an important factor; however, as the existing literature on this topic has contradicting results, interpretation must be done with caution.

PCS fatigue may further explain some of the associations between COVID-19 and relative regional volume. In this study, the subjects > 6 months post-infection who reported PCS fatigue symptoms (i.e., fatigued subjects) showed lower volume in frontal lobe areas (bilateral mid-frontal and straight gyri, left inferior gyrus) compared to people without post-COVID-19 fatigue symptoms (i.e., non-fatigued subjects). Additionally, FSS and FAS scores were associated with lower relative volumes in these same frontal areas. Previous studies have reported conflicting results regarding the effect of fatigue on brain morphometry in people with PCS symptoms. Specifically, Besteher et al. showed increased volumes in frontal areas in people with persistent fatigue after COVID-19 infection, while Bispo et al. found decreased white matter tract density in persistently fatigued subjects, both compared to individuals who never had COVID-19 [13,27]. The decreased density in white matter tracts reported by Bispo et al. included tracts connected to the frontal lobe such as the cingulum and the inferior fronto-occipital fasciculus [27,51,52]. This is critical as it may help explain the decrease in the motor cortex (M1) excitability and hypometabolism that have been reported in PCS patients with fatigue [15,53]. Motor disturbances were also reported in another neurological disorder where fatigue is the chief complaint, specifically, chronic fatigue syndrome, and it has been suggested that it may be due to decreased myelination of white matter tracts connected to M1 [27]. Similarly, fatigue is a common neurological symptom in people with multiple sclerosis, which may be related to decreased myelination and may inform studies on people with PCS fatigue. [54]. In MS patients with fatigue, thalamic and basal nuclei atrophy have been reported, along with demyelination and axonal loss in the thalamus [27]. Furthermore, it has been suggested that inflammation in MS may alter functional connectivity in certain networks, resulting in fatigue [55], and similar chronic inflammation has been reported in PCS patients [56]. Thus, it is possible that a similar mechanism may be occurring in COVID-19 after acute infection. It may be the result of a complex interplay of thalamic and basal nuclei atrophy combined with chronic inflammation and altered M1 physiology that results in the high prevalence of fatigue [6].

Unexpectedly, alterations in brain volume at different time points in PCS subjects were not accompanied by differences in glucose metabolism. Previous studies have shown alterations in only structure or function, but not both, in the COVID-19 population. Specifically, alterations in cerebral glucose metabolism without alterations in volumes have been shown in several PCS populations and a case report of an active COVID-19 infection [57,58,59,60]. Comparison to our study must be done with caution, as the severity of the acute COVID-19 infections in previous studies was different from the subjects in the current study (i.e., severe vs. mild infection). While it is unexpected to have morphometric changes but not metabolic changes, the absence of metabolic changes alone has precedent, as a recent study showed that nearly half of the subjects reporting symptoms around 10 months after acute COVID-19 infection showed no brain metabolic abnormalities [15,61]. In addition, Nugent and colleagues suggested that cortical thinning/atrophy may precede metabolic changes, which have been reported in longitudinal studies on Alzheimer’s disease [62,63,64]. Therefore, it is plausible that the morphometric changes seen here could be preceding the development of functional/metabolic changes.

Interestingly, compared to a commercially available, proprietary normative database (MIM) compiled before the advent of COVID-19 (i.e., no COVID-19 diagnosis possible), subjects < 6 months showed hypometabolism in the pallidum and hypermetabolism in the lateral orbital gyrus. Additionally, subjects > 6 months displayed hypometabolism in the pallidum and hypermetabolism in the caudate nucleus. Importantly, hypometabolism in the caudate nucleus could have an effect on cognition based on reports of a hypometabolic caudate nucleus playing a key role in Parkinson’s disease dementia [65]. This is significant because cognitive impairment is prevalent after COVID-19 infection [66], and cortical hypometabolism was associated with cognitive impairments after PCS [10,17]. Furthermore, these results underscore the importance of comparing studies based on the nature of the control group prior to interpretation. The current study showed different results when comparing to other recovered COVID-19 patients versus comparing to healthy subjects from a normative database with no possible COVID-19 diagnosis. Alterations compared to healthy controls are consistent with previous PCS literature [15,16]. Hypometabolism in PCS subjects is most consistently reported, yet there are some reports of hypermetabolism, especially early after the COVID-19 infection [56]. Interestingly, Hosp et al. interpreted the hypermetabolism as artifacts of “preserved” metabolism (i.e., not actually hypermetabolism), based on the weights in the exploratory analysis, but that interpretation has been called into question [67]. Instead, it was suggested that initial hypermetabolism in the early phases may give way to hypometabolism during PCS progression, suggesting a secondary injury mechanism separate from the acute infection. However, this interpretation is still questioned [68]. While the mechanism is up for debate, the regions that demonstrated significant alterations (i.e., hypo- or hypermetabolism) in previous studies are similar to the regions showing altered metabolism in the current study, such as the orbital, brainstem, and cerebellar areas [67]. The results of the current study would suggest there is a combination of hyper- and hypometabolism at play both initially and >6 months post-infection. However, future studies are required to elucidate these effects.

### Limitations and Future Studies

There are several strengths and limitations of the current study. First, assessing both structural and metabolic brain changes in people after COVID-19 at various time points (i.e., <6 and >6 months post-infection) is a strength of this study. A major limitation of this study is the small sample size relative to other MRI studies, especially since the patient groups are heterogeneous with respect to age and sex, two factors that may influence regional volumes and metabolic rates [12]. However, evaluating the data using normalization schemes that partially account for these factors was intended to help mitigate this. Moreover, although potential subjects were excluded from the study if they reported the presence of a condition known to exacerbate PCS fatigue symptoms, we did not assess additional factors that may influence these symptoms, such as levels of anxiety, depression, and psychological stress, including pre-infection levels of these variables. Other lifestyle factors that may have affected the results, such as pre-infection physical activity levels and sleep quality, were also not addressed. Furthermore, cognition and its associations with imaging outcomes were not assessed, nor was information on the presence of other common PCS symptoms outside of fatigue (e.g., dysosmia or ageusia) provided. Lastly, the potential influences of the COVID-19 variant, treatment with antivirals, and vaccination status were not assessed in this study due to the timing of the recruitment of the subjects and vaccine availability.

Future imaging studies with larger sample sizes are needed to investigate the effects of acute COVID-19 on brain volume and glucose metabolism and to determine the role of these changes in post-COVID-19 fatigue. These studies should evaluate if these changes are specific to the COVID-19 variant that people were infected with and if vaccination and treatment status influence these differences. Additionally, future studies should assess if PCS-related brain changes are affected by the age and sex of subjects, particularly because both women and older subjects report more severe PCS symptoms [4,69]. The role of psychological influences, such as levels of anxiety and depression, as well as lifestyle factors, such as physical activity participation, should also be evaluated. Ideally, longitudinal studies at various time points (e.g., immediately after acute COVID-19 infections until ~1 year after infection) should be conducted to investigate alterations in brain structure and function throughout the duration of PCS symptoms.

## 5. Conclusions

This study found that COVID-19 may affect relative regional volume but not glucose metabolism, particularly in people >6 months post-infection who report PCS fatigue symptoms. Furthermore, both groups (e.g., >6 and <6 months post-infection) showed regional hypo- and hypermetabolism compared to a normative database. This work is highly significant and contributes to the development of the literature on the effects of mild COVID-19 on brain structure and function over time. However, more imaging studies with larger sample sizes and cognitive function measures are required to elucidate these effects and explain the variability currently reported in the PCS literature.

## Figures and Tables

**Figure 1 brainsci-13-00675-f001:**
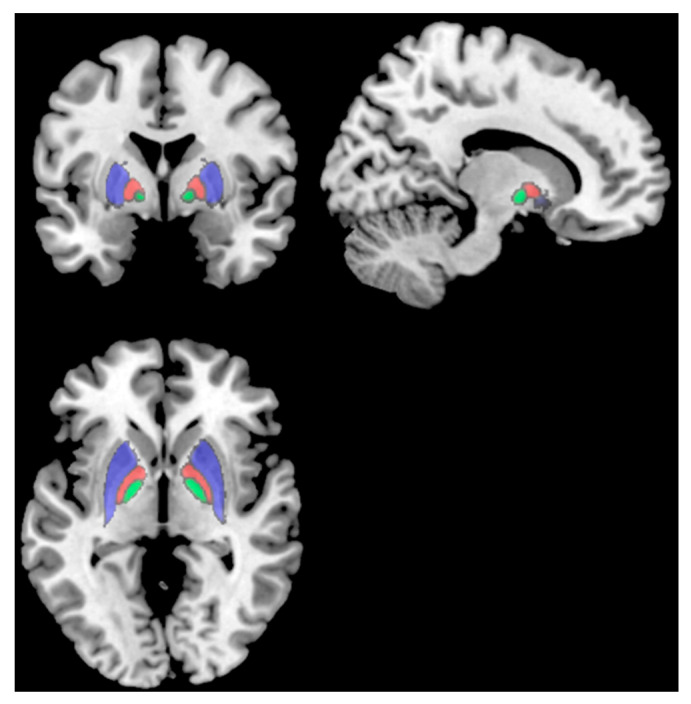
Single subject MNI template [36] T1 MRI showing the putamen (red), pallidum (cyan), and thalamus (green), which all had smaller volumes in subjects > 6 months post-infection than subjects < 6 months post-infection.

**Figure 2 brainsci-13-00675-f002:**
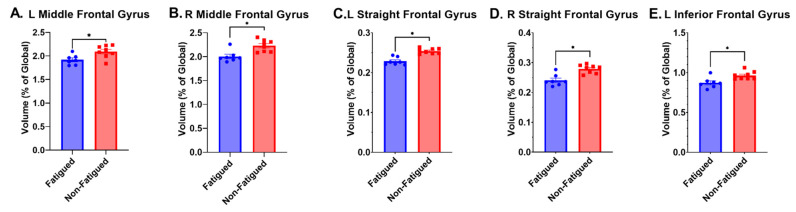
Regional volumetric comparisons for fatigued (blue) and non-fatigued (red) subjects > 6 months post-infection. Y axis is the percent of total volume (i.e., 1.5 = 1.5%). Dots represent individual data points. * Represents a *p* < 0.05. Data are means ± SEM. Fatigued subjects > 6 months post-infection had smaller relative volumes across each brain region.

**Figure 3 brainsci-13-00675-f003:**
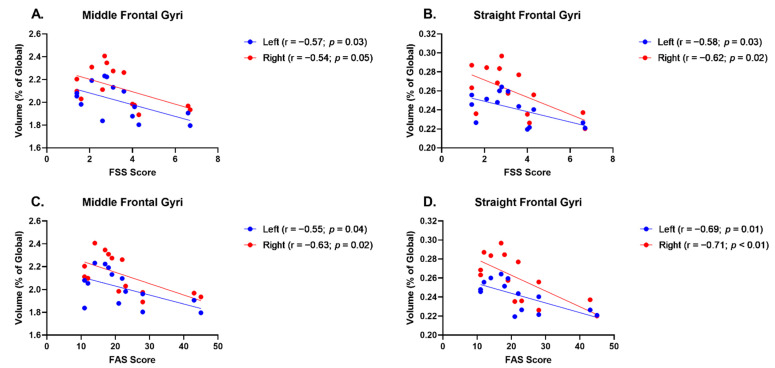
Pearson’s correlations for FSS (**A**,**B**) or FAS (**C**,**D**) and RRV in the middle and straight frontal gyri in subjects > 6 months post-infection. In all cases, a smaller RRV was associated with worse fatigue (FSS) and perceived fatigability (FAS).

**Table 1 brainsci-13-00675-t001:** (**a**). Subject characteristics for the <6 months and >6 months groups. (**b**). Sub-group characteristics and outcome of the FSS and FAS for the fatigued and non-fatigued groups at each time point.

(a). Subject Characteristic for the <6 Months and >6 Months Groups.						
Group		<6 Months Post-Infection		>6 Months Post-Infection		*p*-Value
N (f)		18 (11)		15 (6)		0.49
# Fatigued (female)		9 (6)		7 (2)		0.85
CFQ-11 Score (F)		7.80 ± 2.15		8.57 ± 2.57		0.36
CFQ-11 Score (NF)		1.56 ± 1.58		1.13 ± 1.36		0.41
Age (years)		36.61 ± 15.66		30.40 ± 13.01		0.23
Weight (kg)		78.66 ± 21.79		80.50 ± 22.42		0.81
Height (cm)		167.78 ± 39.81		174.75 ± 44.78		0.64
BMI		28.09 ± 8.08		26.38 ± 7.19		0.53
Time since Infection (months)		3.68 ± 1.69		12.61 ± 4.79		<0.01
# Vaccinated		16		15		
# Vaccinated before infection		10		3		
# Vaccinated after infection		6		12		
**(b). Sub-group characteristics and outcome of the FSS and FAS for the fatigued and non-fatigued groups at each time point.**						
Group	<6 months F	<6 months NF	*p*-Value	>6 months F	>6 months NF	*p*-Value
Age	39.6 ± 16.2	36.6 ± 15.7	0.70	37.1 ± 12.4 (18–54)	24.5 ± 4.6 (19–32)	0.02
N (f)	9 (6)	9 (5)	0.65	7 (2)	8 (4)	0.44
Height (cm)	169.3 ± 11.98	166.2 ± 9.1	0.55	172.7 ± 7.5	176.5 ± 12.3	0.49
Weight (kg)	78.9 ± 11.3	78.7 ± 21.8	0.98	81.8 ± 9.1	79.4 ± 11.6	0.67
FSS Score	4.6 ± 1.5	2.1 ± 0.5	<0.001	4.4 ± 1.8	2.3 ± 0.7	0.009
FAS Score	28.6 ± 7.8	16.0 ± 5.1	<0.001	30.0 ± 10.0	14.6 ± 3.4	0.001

All data are mean +/− sd. Fatigued was defined as a score ≥ 5 on the Chalder Fatigue Scale. Range of age was provided in parentheses for >6months F vs. NF to give further context. F = Fatigued; NF = Non-fatigued. FSS = Fatigue Severity Scale; FAS = Fatigue Assessment Scale; # = Number. *p*-Values represent results of unpaired *t*-test or chi-square test.

**Table 2 brainsci-13-00675-t002:** (**a**). Results of the RRV comparison between the two time points. (**b**). Results of the RRV comparisons between fatigued and non-fatigued at each time point.

(a). Results of the RRV Comparison between the Two Time Points.						
Region	<6 Months Post-Infection		>6 Months Post-Infection	*p*-Value	Cohen’s d	
L Putamen	0.41 ± 0.02		0.43 ± 0.02	0.01	1.00	
R Putamen	0.40 ± 0.03		0.42 ± 0.02	0.03	0.77	
L Pallidum	0.11 ± 0.008		0.12 ± 0.007	0.01	1.30	
R Pallidum	0.11 ± 0.009		0.12 ± 0.007	0.003	1.23	
L Thalamus	0.59 ± 0.04		0.62 ± 0.03	0.03	0.84	
R Thalamus	0.57 ± 0.04		0.60 ± 0.03	0.01	0.84	
R Cuneus	0.59 ± 0.05		0.55 ± 0.04	0.01	0.87	
**(b). Results of the RRV comparisons between fatigued and non-fatigued at each time point.**						
Region	<6 months F	<6 months NF	>6 months F	>6 months NF	*p*-value	Cohen’s d
L Middle Fr. Gyrus	*	*	1.92 ± 0.11	2.09 ± 0.13	0.02	1.40
R Middle Fr. Gyrus	*	*	2.01 ± 0.12	2.23 ± 0.12	0.004	1.83
L Straight Fr. Gyrus	*	*	0.23 ± 0.01	0.25 ± 0.01	<0.001	2.00
R Straight Fr. Gyrus	*	*	0.24 ± 0.02	0.28 ± 0.01	<0.001	2.60
L Ant. Orbital Gyrus	*	*	0.33 ± 0.03	0.37 ± 0.01	0.02	1.85
L Inferior Fr. Gyrus	*	*	0.87 ± 0.07	0.96 ± 0.05	0.01	1.50
L Mid. Orbital Gyrus	*	*	0.33 ± 0.02	0.37 ± 0.02	0.005	2.00
R Mid. Orbital Gyrus	*	*	0.32 ± 0.03	0.36 ± 0.03	0.03	1.33
R Sup. Post. Temp. Gyrus	*	*	0.65 ± 0.06	0.72 ± 0.05	0.04	1.28
R Ant. Cingulate Gyrus	*	*	0.41 ± 0.03	0.45 ± 0.03	0.03	1.33
L Caudate Nucleus	0.32 ± 0.03	0.35 ± 0.02	*	*	0.05	1.19
R Nucleus Accumbens	0.025 ± 0.002	0.027 ± 0.002	*	*	0.04	1.00
L Substantia Nigra	0.028 ± 0.002	0.030 ± 0.002	*	*	0.04	1.00

All data are mean ± SD. *p*-Value is derived from unpaired *t*-test comparisons. * Represents not applicable. Fr = Frontal; Ant = Anterior; Mid = Middle; Post = Posterior; Sup = Superior; Temp = Temporal; F = Fatigued; NF = Non-fatigued.

**Table 3 brainsci-13-00675-t003:** Results of the Pearson’s correlations between FAS or FSS and RRV of specific brain regions in people >6 months post-infection (top) and people <6 months post-infection (bottom). Regions with significant correlations in both fatigue scales are **bolded**, and regions with a significant correlation in only one fatigue scale are *italicized*.

>6 Months Post-Infection				
**Region**	**FSS r-Value**	***p*-Value**	**FAS r-Value**	***p*-Value**
**L Middle Frontal Gyrus**	**−0.57**	**0.03**	**−0.55**	**0.04**
**R Middle Frontal Gyrus**	**−0.54**	**0.05**	**−0.63**	**0.02**
**L Straight Frontal Gyrus**	**−0.58**	**0.03**	**−0.69**	**0.01**
**R Straight Frontal Gyrus**	**−0.62**	**0.02**	**−0.71**	**<0.01**
L Anterior Orbital Gyrus	−0.43	0.13	−0.48	0.09
**L Inferior Frontal Gyrus**	**−0.56**	**0.04**	**−0.65**	**0.02**
*L Middle Orbital Gyrus*	−*0.46*	*0.10*	−*0.55*	*0.04*
*R Middle Orbital Gyrus*	−*0.48*	*0.08*	−*0.65*	*0.01*
R Superior Posterior Temporal Lobe	−0.15	0.61	−0.24	0.40
R Anterior Cingulate Gyrus	−0.41	0.15	−0.54	0.05
**<6 Months Post-Infection**				
Region	FSS r-Value	*p*-Value	FAS r-Value	*p*-Value
L Caudate Nucleus	−0.22	0.40	−0.22	0.40
**R Nucleus Accumbens**	**−0.60**	**0.01**	**−0.51**	**0.04**
L Substantia Nigra	−0.45	0.07	−0.43	0.09

FSS = Fatigue Severity Scale; FAS = Fatigue Assessment Scale.

## Data Availability

The data that support the findings of this study are available on request to the corresponding author.

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
