# Peer review of "Effect of Post-COVID-19 on Brain Volume and Glucose Metabolism: Influence of Time Since Infection and Fatigue Status"

_brainsci, 2023, doi:10.3390/brainsci13040675_

Round 1

Reviewer 1 Report

This study investigates the effect of post-COVID-19 condition on both brain volume assessed by brain MRI and glucose metabolism evaluated by PET. This study is unique in that it measures both brain volume and metabolism although almost no significant correlation were detected. Another unique point is that it separates the patients by the duration of the symptoms since COVID-19 infection (<6M versus >6M). 

Althouth the number of samples is limited and the demography of each cohort is not well balanced, the results contain interesting findings such as brain regions associated with fatigue of post-COVID-19 patients.  

Here is some specific comments from this reviewer.

1. As the authors mention in the discussion, limitation includes vacccine status. In Table 1, "vaccinated before infection" and " vaccinated after infection" seem not well balanced between the 2 cohorts (eg. proportion of sex with fatigued individuals) . The chi-square test should be performed for each. It is advisable to put p values of each analysis (t-tests and chi-square test) in the Table 1.

2. The presentation of each data can be improved (for example, to make each table for each analysis).

3. The readers will be benefited if there is a table summarizing the current results and the ones by previous publications.

Author Response

Thank you very much for your helpful and constructive comments and suggestions.

  1. As the authors mention in the discussion, limitation includes vaccine status. In Table 1, "vaccinated before infection" and " vaccinated after infection" seem not well balanced between the 2 cohorts (eg. proportion of sex with fatigued individuals) . The chi-square test should be performed for each. It is advisable to put p values of each analysis (t-tests and chi-square test) in the Table 1.
  • P-values were added to table 1. Vaccination status was not included in the comparison, and evaluation of vaccine effectiveness for preventing PCS is beyond the scope of this study. Recent work has shown it may play a role, thus we provided this information for further context. A sentence explaining this was added to lines 215-219.
  1. The presentation of each data can be improved (for example, to make each table for each analysis).
  • Tables were added and the results section was edited to make reading more palatable. We thank you for the recommendation, it has made the section much easier to read.
  1. The readers will be benefited if there is a table summarizing the current results and the ones by previous publications.
  • A review outside of the relevant literature is not within the scope of the article. However, a statement referencing a review on this topic has been added to lines 93-94.

Reviewer 2 Report

The manuscript reports a study with MRI and PET in people with a history of COVID infection. The authors aimed to evaluate the differences between early post-covid conditions (less than 6 months) and long periods. The goal is interesting but there are several aspects that reduced my enthusiasm about the manuscript and that required serious consideration by the authors.

- the sample sizes are really small for the goals. How many people responded to your invitation? Who performed the evaluation for inclusion/exclusion criteria? How have you evaluated people's mental and physical conditions before COVID? 

- the authors reported that the mean ages are lower than 40. But the SD is very high, so a lot of people might be over 40 years old. 

- How can you relate the FAS scores to the Covid condition after more than 6 months if you do not have the evaluation before COVID?

- You should use a correction for the p-values due to the small samples, the numbers of the analyses, and the use of neuroimaging data (where is the standard).

- It is not clear to me the reasons for the evaluation of the COVID vaccination. 

- have you evaluated the lateralizations of the participants?

Author Response

We thank the reviewer for the helpful and constructive comments and suggestions.

- the sample sizes are really small for the goals. How many people responded to your invitation? Who performed the evaluation for inclusion/exclusion criteria? How have you evaluated people's mental and physical conditions before COVID? 

  • Information about the recruiting process was added to lines 110-112. Pre-infection health could be a key contributor, and further clarification was added to the limitations sections since we did not assess this (lines 408-409).

- the authors reported that the mean ages are lower than 40. But the SD is very high, so a lot of people might be over 40 years old. 

  • Range of age was added to table 2b and context added to lines 213-215.

- How can you relate the FAS scores to the Covid condition after more than 6 months if you do not have the evaluation before COVID?

  • This is a main limitation of our study, as it is not a longitudinal study but a cross-sectional study. However, it is a preliminary investigation of different time points designed to inform future studies which will be longitudinal in nature and investigate this question.

- You should use a correction for the p-values due to the small samples, the numbers of the analyses, and the use of neuroimaging data (where is the standard).

  • This is an exploratory study with a focused hypothesis on regions to investigate and thus we limited the comparisons to only brain regions shown to have differences in the literature already. Additionally, we have published similar work which has a similar protocol, therefore we felt it was appropriate to not correct for multiple comparisons.

- It is not clear to me the reasons for the evaluation of the COVID vaccination. 

  • Clarification for the inclusion of this information was added to lines 215-219.

- have you evaluated the lateralizations of the participants?

  • Lateralization was evaluated by splitting the regions into right and left, however no consistent patterns emerged.

Round 2

Reviewer 2 Report

The manuscript has improved the information needed for replication. The study maintains some weaknesses, but these are not compromising the validity of the results or the discussion because the authors reported clearly their limits.